# Comparison of Safety of Different Vaccine Boosters Following Two-Dose Inactivated Vaccines: A Parallel Controlled Prospective Study

**DOI:** 10.3390/vaccines10040622

**Published:** 2022-04-15

**Authors:** Zhi-Qiang Lin, Jiang-Nan Wu, Rong-Dong Huang, Fang-Qin Xie, Jun-Rong Li, Kui-Cheng Zheng, Dong-Juan Zhang

**Affiliations:** 1Fujian Provincial Center for Disease Control and Prevention, Fuzhou 350012, China; zhiqianglin2022@163.com (Z.-Q.L.); huangrd1989@163.com (R.-D.H.); okxfq@163.com (F.-Q.X.); junrong_li@126.com (J.-R.L.); kingdadi9909@126.com (K.-C.Z.); 2Obstetrics and Gynecology Hospital, Fudan University, Shanghai 200012, China; wjnhmm@126.com

**Keywords:** booster vaccination, safety, adenovirus type-5 vectored vaccine, protein subunit vaccine, inactivated vaccine

## Abstract

A vaccine booster to maintain high antibody levels and provide effective protection against COVID-19 has been recommended. However, little is known about the safety of a booster for different vaccines. We conducted a parallel controlled prospective study to compare the safety of a booster usingfour common vaccines in China. In total, 320 eligible participants who had received two doses of an inactivated vaccine were equally allocated to receive a booster of the same vaccine (Group A), a different inactivated vaccine (Group B), an adenovirus type-5 vectored vaccine (Group C), or a protein subunit vaccine (Group D). A higher risk of adverse reactions, observed up to 28 days after injection, was found in Groups C and D, compared to Group A, with odds ratios (OR) of 11.63 (95% confidence interval (CI): 4.22–32.05) and 4.38 (1.53–12.56), respectively. Recipients in Group C were more likely to report ≥two reactions (OR = 29.18, 95% CI: 3.70–229.82), and had a higher risk of injection site pain, dizziness, and fatigue. A gender and age disparity in the risk of adverse reactions was identified. Despite the majority of reactions being mild, heterologous booster strategies do increase the risk of adverse reactions, relative to homologous boosters, in subjects who have had two doses of inactive vaccine.

## 1. Introduction

Vaccination can help build herd immunity and provide high efficacy against coronavirus disease 2019 (COVID-19). However, it has been reported that the efficacy wanes, along with a decrease in antibody titer, after the second dose inoculation of multiple types of vaccines [1]. For example, the efficacy of BNT162b2 against COVID-19 waned from 95% between 7 days and 2 months to 84% between 4 and 6 months after the second dose [2,3]. Despite the effectiveness of the vaccines against severe disease, hospitalization and death remain high, thereforea booster was recommended [4].

Homologous boosters of several vaccines have been identified to increase neutralizing antibody titers and provide high efficacy [4,5,6]. However, in the context of increasing vaccine types and adequate supply, a vaccine booster might have better operability and ability to improve the coverage of a booster if there is no restriction on the type of vaccine (e.g., comparable effectiveness and tolerable safety). In addition, heterologous prime-booster strategies may offer an immunologic advantage to extend the breadth and longevity of protection provided by the currently available vaccines [7]. Therefore, using a different vaccine booster might be more beneficial than boosting with the same vaccine. Indeed, previous studies have demonstrated that administration of a different type of vaccine boost induced a robust immune responsewith an acceptable and manageable reactogenicity profile [1,8]. Some previous studies have identified that an mRNA vaccine booster was more immunogenic than a two-dose homologous ChAdOx1 vaccine regimen [9,10,11]. However, in China, little is known about adverse reactions after a booster with either the same or different technical vaccine route. In China, an inactivated vaccine manufactured by Beijing Institute of Biological Products was first approved on 30 December 2020 [12], followed by the Sinovac inactive vaccine on 5 February 2021 [13]. Thereafter, increasingnumbers of vaccines became available, including the CanSino adenovirus type-5 vectored vaccine andthe Zhifei protein subunit vaccine [14,15]. We conducted a parallel controlled prospective study among those who had received two doses of inactivated vaccines, to estimate the safety of a booster using these four common vaccines.

## 2. Methods

### 2.1. Study Design and Data Collection

This study was a paralleled controlled prospective study conducted in ZhangpingCounty, Fujian Province, between August 2021 and January 2022. For eligible subjects, baseline information (such as date of birth, gender, height, weight, chronic disease, and previous vaccination information) was collected. A physical examination, including blood pressureand body temperature measurements, and a general physical examination, as well as a urine pregnancy test, was performed. Data relating to the booster vaccine, such as vaccination date, type of vaccine, and adverse reactions were obtained. This study was approved by the ethics committee of the Fujian Center for Disease Control and Prevention. Written informed consent was signed by all participants.

### 2.2. Inclusion and Exclusion Criteria

Inclusion criteria for the present study included local residents who: (1) were aged 18–59 years; (2) had received 2 doses of inactivated vaccine; (3) had an interval of ≥6 months since the second dose; and (4) had good physical and mental health. Those with a clear history of COVID-19 infection, who used blood products or immunosuppressants after the previous dose of the vaccine, or had allergic or other abnormal reactions (e.g., high fever ≥ 39.0 °C or serious nervous system reactions) after the second dose were excluded. Pregnant women and those who suffered an acute disease were also excluded from the study. 

### 2.3. Booster Groups

After beingmade aware of the four booster vaccines, including the inactivated vaccine produced by Beijing Institute of Biological Products; Sinovac, an adenovirus type-5 vectored vaccine manufactured by CanSinoBiogics; and the protein subunit vaccine from AuhuiZhifeiLongcom Biopharmaceutical, participants independently selected one vaccine as a booster. For BBIBP-CorV, the 19nCoV-CDC-Tan-HB02 strain was purified and passaged in Vero cells to generate the stock for vaccine production. The stock virus was replicated over 7.0 log10 cell cultures, had an infectious dose 50% assay by 48–72 h post infection, a multiplicity of infection of 0.01–0.3, and was inactivated by mixing β-propionolactone with the harvested viral solution [16]. Sinovac-CoronaVacwas created from Vero cells that had been inoculated with the CN02 strain of severe acute respiratory syndrome coronavirus 2 and inactivated with β-propiolactone [17]. An optimized full-length spike gene was cloned based on Wuhan-Hu-1 (GenBank accession number YP_009724390) with the tissue plasminogen activator signal peptide gene inserted into an E1 and E3 deleted adenovirus type-5 vector, to construct the adenovirus type-5 vectored COVID-19 vaccine expressing the spike glycoprotein of severe acute respiratory syndrome coronavirus 2 [18]. The ZF2001 protein subunit vaccine was developed using a dimeric form of the receptor-binding domain of the severe acute respiratory syndrome coronavirus 2 spike protein as the antigen (residues 319–537, accession number YP 009724390) in the CHOZN CHO K1 cell line [19].

We stopped the recruitment for each group when the target of 80 subjects was satisfied. According to the study design, four groups were set up: Group A received the same brand of inactivated vaccine as their previous two-dose vaccine; Group B chose a brand of inactivated vaccine different from their former two-dose inactivated vaccine; Group Chad anadenovirus type-5 vectored vaccine as the booster; and Group D received aprotein subunit vaccine booster.

### 2.4. Outcomes

Adverse reactions in the present study were defined as reactions related to the booster vaccination. Data on suspected abnormal reactions were collected by the staff or reported by the subjects up to the 28th day after vaccination. Adverse reactions included injection site, systemic, or other reactions, with injection site reactions referring to indurations, redness, and swelling. Systemic reactions included allergy, vomiting, diarrhea, rash, cough, etc. All reported adverse reactions were recorded and handled in accordance with the national monitoring program for adverse reactions [20].

### 2.5. Data Analysis

We calculated the proportion and 95% confidence intervals (95% CI) of adverse reactions in each group.Categorical variables were compared across the groups using the chi-squared or Fisher’s exact test, while the ANOVA method was used for continuous variables.The Bonferroni method was used for pairwise comparisons across the four groups if statistical significance was found.

Logistic regression analysis was conducted relative to the group with the same brand of inactivated vaccine booster, to estimate the risk for total adverse reactions in the other groups. Odds ratios (ORs) and corresponding 95% confidence intervals (CI) were calculated. We also constructed a multinomial logistic regression model to compare the risk of 1 and ≥2 reactions after the booster among subjects in the four groups. In the adjusted model, we included potential confounders such as age (years), gender (male or female), body mass index (BMI, calculated from weight and height, kg/m^2^), and chronic disease (yes or no). A Kaplan–Meier curve was constructed to compare the cumulative risk of adverse reactions across the four groups.Specific reactions across the four groups were also compared using the chi-squared or Fisher’s exact test, and the Bonferroni method if applicable.

The receiver operating characteristic (ROC) method was run to identify a cut-off value for further classification of a continuous variable (e.g., age). Stratification analysis was utilized to compare the difference in adverse reaction risk for category variables significantly related to adverse reactions (e.g., male vs. female; age ≥ vs. <41 years old). All analyses were performed using SPSS version 23.0 (IBM corp., Armonk, NY, USA). A two-sided *p*-value of <0.05 was regarded as statistically significant.

## 3. Results

### 3.1. Basic Characteristics

A total of 320 eligible subjects were recruited and assigned to the four groups (Figure 1). Among these participants, 192 (60.0%) were male, and 17 (5.3%) had chronic disease. The subjects were aged from 20 to 58 years, with a mean of 43.2 years (SD = 9.0). The mean BMI was 23.8 kg/m^2^ (SD = 4.1). Basic characteristics were balanced across the four groups (Table 1). After the booster, 64 subjects reported at least one adverse reaction, with a reporting rate of 20.0% (95% CI: 15.6–24.4%). Most adverse events were mild. Among them, eight participants (2.5%, 95% CI: 0.8–4.2%) observed both injection site reaction and systemic outcomes, and one (0.3%, 95% CI: 0–0.9%) reported both systemic and other adverse reactions. An injection site adverse reaction was observed by 46 individuals (14.4%, 95% CI: 10.5–18.2%), 24 (7.5%, 95% CI: 4.6–10.4%) reported a systemic outcome, and 3 (0.9%, 95% CI: 0–2.0%) emerged with other adverse reactions.

### 3.2. Risk of Adverse Reactions across the Four Groups

After the vaccination, 5, 5, 36, and 18 participants fromgroups A, B, C, and D, respectively, observed adverse reactions, with reported adverse reaction ratesat 6.2%, 6.2%, 45.0%, and 22.5%, respectively (*p* < 0.001, Table 1). Logistic regression analysis indicated that, compared with subjects in group A, those having an adenovirus type-5 vectored vaccine booster had the highest risk of adverse outcomes, with an adjusted OR of 11.63 (95% CI: 4.22–32.05). People who were givena protein subunit vaccine booster also had a higher risk of reaction (adjusted OR = 4.38, 95% CI: 1.53–12.56). However, there was no significant difference in the adverse reaction rate between individuals in groups A and B (Table 2).

Similar results were found for injection site adverse reactions across the four groups. The corresponding adjusted ORs were 12.58 and 5.18 for groups C and D, respectively (Table 3). For systemic adverse outcomes, a higher risk was identified in Groups C and D. However, no significant association was found in the recipients in Group D (adjusted OR = 6.97, 95% CI: 0.81–60.17, *p* = 0.078) (Table 4). In addition, being female was associated with an increased risk of injection site adverse reactions, relative to being male (Table 3). Age was negatively correlated with systemic reactions (Table 4).

Common specific reactions at the injection site were pain and swelling, while dizziness and fatigue were common systemic reactions. There were significant differences in these four specific reactions across the four groups (*p* ≤ 0.01) (Table 5). In the pairwise comparisons by the Bonferroni method, subjects in Group C reported a higher rate of injection site pain, dizziness, and fatigue than those in groups A or B. The risk of dizziness in group C was also higher relative to that of group D (Table 5). No difference was found between other groups.

### 3.3. Number and Cumulative Risk of Reactions

Multinomial logistic regression analysis indicated that, compared with subjects in group B, the risk of having one reaction after the booster was higher in groups C and D, with an adjusted OR of 7.8 (95% CI: 2.46–24.68) and 4.69 (95% CI: 1.45–15.17), respectively (Table 6). Participants in group C also had an increased likelihood of reporting ≥two adverse reactions relative to those in group B. However, these risks for group D participants did not differ significantly from those in group B (Table 6). The cumulative risk of total, injection site, and systemic reactions varied across the four groups (all log-rank *p* values < 0.001) (Figure 2).

### 3.4. Stratification Analysis

Stratification analysis indicated that, compared with those having an inactivated vaccine booster, the risk of reactions at the injection site in female individuals receiving theadenovirus type-5 vectored vaccine booster was significantly increased, and differed from that in male participants (OR: 15.26 (5.21–44.68) vs. 2.89 (0.49–17.19)) (Table 7). Age had a mild-to-moderate predictive value for no reaction in the ROC method, with an area under the curve of 0.65 (95% CI: 0.54–0.76, *p* = 0.013). The cut-off value of the model was 41 years old, with a corresponding sensitivity of 65.2% and a specificity of 62.5%. Similar age group disparity (≥ vs. <41 years) for systemic reactions in recipients of the protein subunit vaccinebooster was found in the stratification analysis (Table 8).

## 4. Discussion

In this parallel controlled prospective study, we found that participants who received an adenovirus type-5 vectored or protein subunit vaccine booster had a higher risk of adverse reactions (e.g., total, injection site, or systemic reactions) than those injected with an inactivated vaccine booster. These risks were particularly obvious in the population receiving an adenovirus type-5 vectored vaccine, manifesting as a significantly higher risk of reporting ≥two adverse reactions, and some specific reactions such as pain at the injection site, dizziness, and fatigue. In addition, a gender disparity for injection site adverse reactions caused by an adenovirus type-5 vectored booster was identified, as well as an age difference (≥ vs. <41 years) in systemic reactions reported in protein subunit vaccine recipients. 

Vaccination may help build herd immunity against COVID-19 [1]. However, the antibody titer decreases over time, which results in a waning protective efficacy. Therefore, a third booster has been proposed. Booster vaccination with BNT162b2 was initiated in Israel to manage cases of COVID-19caused by the delta variant [21,22]. Previous studies have also identified the immunogenicity of a booster given 6–8 months after the second dose of BNT162b2, mRNA-1273, NVX-CoV2373, CoronaVac, and protein subunit (ZF2001) vaccines [1,4,23,24]. Homologous and heterologous booster vaccines were identified as having acceptable safety and comparable immunogenicity in the United States, where three main candidate vaccines (mRNA, Ad26.COV2.S, and BNT162b2) were approved [7]. On 19 November 2021, the US Food and Drug Administration approved a BNT162b2 or mRNA vaccine booster for people aged ≥18 years old. 

In China, inactivated vaccines were approved first. Thus, the vast majority of residents were vaccinated with two-dose inactive vaccines. More vaccines have been approved since then, and concerns about the immunity and safety of sequential boosters using different vaccines have arisen. There would be more choice for clinic staff and residents if the adverse reaction risks were tolerable and the efficacy was competitive. This might further improve the coverage of booster vaccinations.

The safety of the COVID-19 vaccine has been widely evaluated, including monitoring of vaccine recipients for facial paralysis and unusual thrombotic events [25,26]. In the present study, we focused on the adverse reactions of boosters related to the four common vaccines in China. We found that administering a booster of a different technical route increased the risk of adverse reactions relative to using a booster vaccine of the same technical route. In previous clinical studies, adenovirus type-5 vectored or protein subunit vaccine recipients indeed reported a higherrisk of adverse reactions than those receiving inactivated vaccines [17,18,19]. The mechanism of these additional risks of adverse reactions remains unknown, but might be related to the different vaccine-elicited spike-specific CD4+ and CD8+ T-cell responses. These responses may contribute to the durability of the antibody response and to prevention of severe disease [27,28]. Heterologous boosters may provide an immune response that might be beneficial for durability by increasing spike-specific CD8+ T cells [7], resulting in an additional risk of adverse reactions. Our findings might indicate that having aprevious two-dose inactivated vaccine does not decrease the high risk of adverse reactions of these vaccines. Further studies on the reasons for thisare warranted, even considering that the vast majority of the reactions were mild and these high risks were thought to be tolerable. 

In addition, being female was associated with a higher risk of reaction at the injection site for adenovirus type-5 vectored vaccine recipients, while individuals aged < 41 years old had an increased risk of systemic reactions when receiving a protein subunit vaccine booster. We found that subjects aged ≥ 41 years old reported lower rates of adverse reactions. This is interesting, since older age is a risk factor for severe COVID-19 in the general population and for those with autoimmune diseases [29]. However, a better tolerance was found in older adults than in younger adults [30]. This disparity might be related to aging-induced immunosenescence (i.e., the gradual deterioration and decline of the immune system), in which age-dependent differences in the functionality and availability of T-cell and B-cell populations are thought to have a critical role [30,31]. Further studies on the cause of the gender disparity related tothe risk of injection site adverse reaction in subjects who received adenovirus type-5 vectored vaccine are warranted. Nevertheless, these novel findings may provide evidence for guiding the selection of vaccines for booster vaccination. For example, an adenovirus type-5 vectored vaccine may be an alternative for men, and a protein subunit vaccine is also a feasible choice for older people (e.g., ≥41 years old), when an inactivated vaccine booster is not available.

The advantage of the present study is that we are the first to compare the safety of a booster using four common vaccines available in China. The study design and the sample size of 320 may ensure high credibility and robustness of the results. However, there are certain limitations. First, this was not a randomly controlled trial. Despite the characteristics being balanced across the groups, the evidence level of the present study is lower than that of an RCT. Second, the safety of the vaccines in specific populations (e.g., autoimmune patients) was unknown in the present study since we excluded them during recruitment. Third, we discuss the differences across the four groups assuming that their antibody titer and protective efficacy levels were comparable. The applicability of the findings might be affected if this assumption is not tenable.

## 5. Conclusions

In conclusion, for subjects who have received two doses of inactivated vaccine, a booster of a different technical route (i.e., adenovirus type-5 vectored or protein subunit vaccine) was associated with an increased risk of adverse reactions compared with a vaccine booster of the same technical route (i.e., inactivated vaccine). Adenovirus type-5 vectored or protein subunit vaccines might be an alternative for some specific groups, considering that the adverse reactions are mild and tolerable, and given the existing gender and age disparities.

## Figures and Tables

**Figure 1 vaccines-10-00622-f001:**
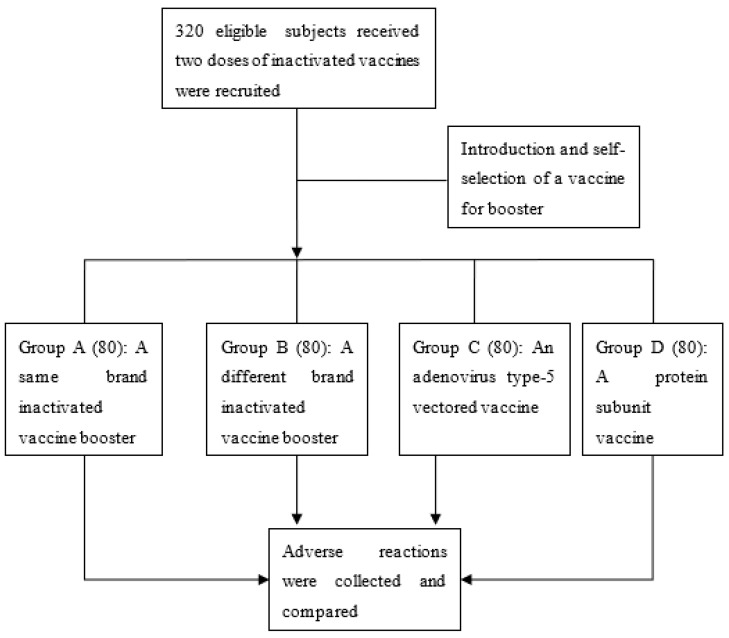
Flow chart of the study.

**Figure 2 vaccines-10-00622-f002:**
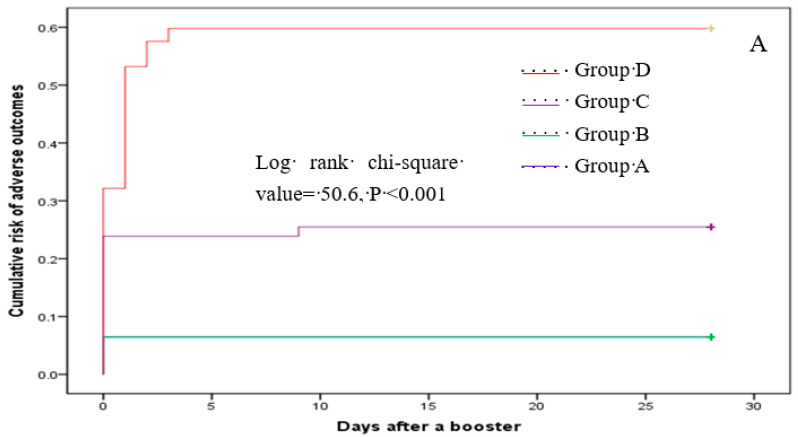
Cumulative risk of total (**A**), local (**B**), and systemic (**C**) reactions.

**Table 1 vaccines-10-00622-t001:** Basic characteristics and adverse reactions after a vaccine booster.

	A (*n* = 80)	B (*n* = 80)	C (*n* = 80)	D (*n* = 80)	*p* Value
Gender					0.72
Male	30 (37.5)	35 (43.8)	29 (36.3)	34 (42.5)	
Female	50 (62.5)	45 (56.2)	51 (63.7)	46 (57.5)	
Chronic diseases					0.20
No	73 (91.3)	79 (98.8)	75 (93.8)	76 (95.0)	
Yes	7 (8.7)	1 (1.2)	5 (6.2)	4 (5.0)	
Age (year), mean (SD)	43.7 (9.5)	42.9 (9.3)	41.3 (8.9)	44.8 (7.8)	0.09
BMI (kg/m^2^), mean (SD)	23.2 (4.0)	23.8 (3.4)	24.1 (5.5)	24.0 (3.1)	0.50
Adverse reactions					<0.001
No	75 (93.8)	75 (93.8)	44 (55.0)	62 (77.5)	
Yes	5 (6.2)	5 (6.2)	36 (45.0)	18 (22.5)	

**Table 2 vaccines-10-00622-t002:** Odds ratio for risk of adverse reactions after a vaccine booster.

	B	S.E.	Wald	*p* Value	OR	95% CI	
						Low	Upper
Groups							
A					1.00		
B	−0.05	0.66	0.01	0.94	0.95	0.26	3.43
C	2.45	0.52	22.51	<0.001	11.63	4.22	32.05
D	1.48	0.54	7.54	0.006	4.38	1.53	12.56
Gender							
Male					1.00		
Female	0.35	0.32	1.19	0.27	1.42	0.76	2.68
Chronic diseases							
No					1.00		
Yes	−0.76	0.81	0.89	0.35	0.47	0.10	2.28
Age (years)	−0.02	0.02	0.78	0.38	0.98	0.95	1.02
BMI (kg/m^2^)	0.03	0.04	0.88	0.35	1.03	0.96	1.11

**Table 3 vaccines-10-00622-t003:** Odds ratio for risk of injection site reactions after a vaccine booster.

	B	S.E.	Wald	*p* Value	OR	95% CI	
						Low	Upper
Groups							
A					1.00		
B	0.60	0.75	0.62	0.43	1.81	0.41	7.94
C	2.53	0.64	15.45	<0.001	12.58	3.56	44.49
D	1.65	0.67	6.10	0.014	5.18	1.40	19.14
Gender							
Male					1.00		
Female	0.74	0.38	3.91	0.048	2.10	1.01	4.38
Chronic diseases							
No					1.00		
Yes	−0.22	0.82	0.07	0.79	0.80	0.16	4.03
Age (years)	0.02	0.02	0.80	0.37	1.02	0.98	1.06
BMI (kg/m^2^)	−0.01	0.04	0.10	0.75	0.99	0.92	1.07

**Table 4 vaccines-10-00622-t004:** Odds ratio for risk of systemic reactions after a vaccine booster.

	B	S.E.	Wald	*p* Value	OR	95% CI	
						Low	Upper
Groups							
A							
B	−0.06	1.43	0.01	0.97	0.95	0.06	15.50
C	2.89	1.05	7.59	0.006	18.00	2.30	140.72
D	1.94	1.10	3.12	0.078	6.97	0.81	60.17
Gender							
Male					1.00		
Female	−0.17	0.46	0.14	0.71	0.843	0.34	2.07
Age (years)	−0.06	0.03	4.38	0.04	0.95	0.90	1.00
BMI (kg/m^2^)	0.05	0.04	1.22	0.27	1.05	0.96	1.14

**Table 5 vaccines-10-00622-t005:** Specific reactions across the four groups.

	A (*n* = 80)	B (*n* = 80)	C (*n* = 80)	D (*n* = 80)	*p* Value
Local reactions	3 ^a^	5 ^a^	25 ^b^	13 ^a,b^	<0.001
Pain	3 ^a^	5 ^a^	25 ^b^	13 ^a,b^	<0.001
Scleroma	0	0	1	0	0.39
Swelling	0	0	6	1	0.01
Blush	0	0	2	0	0.11
Pruritus	0	0	2	0	0.11
Systemic reactions	1 ^a^	1 ^a^	16 ^b^	6 ^a,b^	<0.001
Dizziness	1 ^a^	1 ^a^	12 ^b^	2 ^a^	<0.001
Headache	0	0	3	1	0.11
Cough	0	0	0	1	0.39
Fatigue	0 ^a^	1 ^a,b^	9 ^b^	1 ^a,b^	<0.001
Nausea	0	0	1	1	0.57
Chest tightness	0	0	1	1	0.57
Diarrhoea	0	0	0	1	0.39
Anorexia	0	0	1	0	0.39
Muscle pain	0	0	1	0	0.39
Arthralgia	0	0	0	1	0.39
Other reactions *	1	0	1	1	0.80

Bonferroni method adjusted *p* value > 0.05 for the scores between birth cohortsare marked with the same letter (e.g., groups marked between ^a^ and ^a^, ^b^ and ^b^), otherwise *p* < 0.05 (e.g., groups marked between ^a^ and ^b^). * Other reactions refer to gastrorrhagia in group A, cold symptoms in group C, and sore throat in group D.

**Table 6 vaccines-10-00622-t006:** Odds ratios for risk of number of reactions relative to those without any reaction, multinomial logistic regression model.

No. of Reactions	Groups	B	S.E.	Wald	*p* Value	OR	95% CI	
							Low	Upper
1	B					1.00		
	A	0.32	0.70	0.212	0.65	1.38	0.35	5.39
	C	2.05	0.59	12.20	<0.001	7.80	2.46	24.68
	D	1.55	0.60	6.66	0.01	4.69	1.45	15.17
≥2	B					1.00		
	A					--		
	C	3.37	1.05	10.26	0.001	29.18	3.70	229.82
	D	1.57	1.14	1.90	0.17	4.80	0.52	44.56

Covariates were included in the model, including age (years), gender (male or female), BMI (kg/m^2^), and chronic disease (yes or no).

**Table 7 vaccines-10-00622-t007:** Injection site reaction risk across the four groups, stratified by gender.

	Male (*n* = 128)		Female (*n* = 192)	
	OR (95% CI)	*p* Value	OR (95% CI)	*p* Value
Groups				
A + B	1.00		1.00	
C	2.89 (0.49–17.19)	0.240	15.26 (5.21–44.68)	<0.001
D	5.04 (1.05–24.17)	0.043	3.43 (1.01–11.58)	0.048

Covariates were included in the model, including age (years), BMI (kg/m^2^), and chronic disease (yes or no).

**Table 8 vaccines-10-00622-t008:** Systemic reaction risk across the four groups by age group (≥ vs. <41 years).

	Age ≥ 41 Years (*n* = 209)		Age < 41 Years (*n* = 111)	
	OR (95% CI)	*p* Value	OR (95% CI)	*p* Value
Groups				
A + B	1.00		1.00	
C	21.91 (2.60–184.41)	0.005	22.09 (2.09–233.34)	0.01
D	3.84 (0.33–43.56)	0.280	11.25 (1.05–119.94)	0.045

Covariates were included in the model, including gender (male or female), BMI (kg/m^2^), and chronic disease (yes or no).

## Data Availability

The datasets used in the present study are available from the corresponding author (zhangdj@fjcdc.com.cn) on reasonable request.

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
