# Peer review of "Comparison of Safety of Different Vaccine Boosters Following Two-Dose Inactivated Vaccines: A Parallel Controlled Prospective Study"

_vaccines, 2022, doi:10.3390/vaccines10040622_

Round 1
Reviewer 1 Report
The manuscript that I reviewed “Comparison of safety of different vaccine boosters following two-dose inactivated vaccines: a parallel controlled prospective study” is a study aimed to compare the safety of a booster of four common vaccines in China. 320 patients vaccinated with two doses of inactivated vaccine were divided in four groups, each receiving a booster of the same (Group A) or a different inactivated vaccine (Group B), an adenovirus type-5 vectored vaccine (Group C), or a protein subunit vaccine (Group D). The Authors found that patients who received adenovirus type-5 vectored or protein subunit vaccine booster had a higher risk of adverse reactions than those injected with inactivated vaccines booster. Furthermore, it was observed that female was associated with a higher risk of reactions at injection site for adenovirus type-5 vectored vaccine while individuals aged < 41 years old had increased risk of systemic reactions when receiving a protein subunit vaccine booster.
Comments
As the Author discussed this study is not a random control trial and the principal weakness is the lacking of information about the antibody level of enrolled patients. However, the results obtained are interesting. The article is well written and the statements described are supported by presented data. I have only some observations:
1)Line 111-112: I suggest to the Authors to detail the prevalence as this sentence “Most of the participants were male (192, 60.0%) and 5.3% of them (17) had a chronic disease.” is not clear. “Most of them” seems to refer to male patient and not to all (5.3%, 17/320).
2)Line 115-117: “Among them, 46 individuals observed injection site adverse reaction and 24 reported a systemic outcome, with a proportion of 14.0% (95% CI: 11.0%-18.0%) and 8.0% (95% CI: 5.0%-10.0%), respectively.” I suggest to the Authors to be more clear indicating also the number of patients that observed both the injection site adverse reaction and systemic outcome and I suggest to detail with number the proportion of 14% and 8%.
3)The Authors should deeply discuss the higher risk observed for female and patients < 41 years old.
Author Response
Comments
As the Author discussed this study is not a random control trial and the principal weakness is the lacking of information about the antibody level of enrolled patients. However, the results obtained are interesting. The article is well written and the statements described are supported by presented data.
Author response
Thank you very much for your positive comments. According to the protocol of the study, we would measure the antibody level of enrolled patients at five time-points (e.g., 1 day before the booster, 14 days after the booster, as well as 1, 3, and 6 months after the vaccination, respectively). The final serum sample will be collected between May and June, 2022. We will compare these data when the program is completed.
I have only some observations:
1) Line 111-112: I suggest to the Authors to detail the prevalence as this sentence “Most of the participants were male (192, 60.0%) and 5.3% of them (17) had a chronic disease.” is not clear. “Most of them” seems to refer to male patient and not to all (5.3%, 17/320).
Author response
We have corrected this sentence in the revision. Thank you.
2) Line 115-117: “Among them, 46 individuals observed injection site adverse reaction and 24 reported a systemic outcome, with a proportion of 14.0% (95% CI: 11.0%-18.0%) and 8.0% (95% CI: 5.0%-10.0%), respectively.” I suggest to the Authors to be more clear indicating also the number of patients that observed both the injection site adverse reaction and systemic outcome and I suggest to detail with number the proportion of 14% and 8%.
Author response
Thank you for your reminders. We have described more details on these adverse reactions in result section.
3) The Authors should deeply discuss the higher risk observed for female and patients < 41 years old.
Author response
Thank you. We have discussed more on the novel findings related to female and participants < 41 years old. Please see them in the discuss section.
Reviewer 2 Report
Given the on-going nature of the COVID-19 global pandemic, increased vaccination is an imperative. With a multitude of vaccines available in the marketplace, understanding how different vaccine formulations work in conjunction with each other will be valuable to health care providers. In their paper “Comparison of safety of different vaccine boosters following 1 two-dose inactivated vaccines: a parallel controlled prospective 2 study” Lin et al examine the rate of adverse reactions in a small cohort of patients who had received 2 doses of inactivated vaccines and were boosted using one of 4 possible vaccines. They noted a statistically significant increase in the chance of adverse reactions in patients receiving either a adenoviral based booster or protein subunit booster. They also note that females were more likely to experience adverse advents in response to adenoviral-based booster administration and a potential for adverse reactions in younger individuals to a protein subunit booster. These results will provide valuable information to clinicians and public health officials.
A few concerns will need to be addressed prior to publication:
- More descriptive information about the vaccine products should be provided, apart from the manufacturer. For instance, “protein subunit vaccine” should include the identity of the protein used, which I assume to be viral spike protein, as well as the strain that was used to make it. Similar for the Adeno-V vector and the “inactivated vaccine” (how was it grown, inactivated, strain, etc.).
- I would stress (in multiple points of the paper, such as abstract and discussion) that most adverse reactions were mild (which it appears from table 5).
- More references are needed to round out the paper as several studies have looked at the effect of boosters on antibody titers, overall effectiveness, and adverse events. While the authors are focusing on vaccination primarily in China, reports from other countries with similar products should be included in the introduction and discussion, especially as it relates to the likelihood of adverse events.
Author Response
Given the on-going nature of the COVID-19 global pandemic, increased vaccination is an imperative. With a multitude of vaccines available in the marketplace, understanding how different vaccine formulations work in conjunction with each other will be valuable to health care providers. In their paper “Comparison of safety of different vaccine boosters following 1 two-dose inactivated vaccines: a parallel controlled prospective 2 study” Lin et al examine the rate of adverse reactions in a small cohort of patients who had received 2 doses of inactivated vaccines and were boosted using one of 4 possible vaccines. They noted a statistically significant increase in the chance of adverse reactions in patients receiving either a adenoviral based booster or protein subunit booster. They also note that females were more likely to experience adverse advents in response to adenoviral-based booster administration and a potential for adverse reactions in younger individuals to a protein subunit booster. These results will provide valuable information to clinicians and public health officials.
Author response
Thank you very much for your kindly comments.
A few concerns will need to be addressed prior to publication:
- More descriptive information about the vaccine products should be provided, apart from the manufacturer. For instance, “protein subunit vaccine” should include the identity of the protein used, which I assume to be viral spike protein, as well as the strain that was used to make it. Similar for the Adeno-V vector and the “inactivated vaccine” (how was it grown, inactivated, strain, etc.).
Author response
Thank you for your suggestions. We have provided more information about the vaccine products. Please see them in the method section.
- I would stress (in multiple points of the paper, such as abstract and discussion) that most adverse reactions were mild (which it appears from table 5).
Author response
We mention this information in the revision (e.g., in the abstract and discussion section).
- More references are needed to round out the paper as several studies have looked at the effect of boosters on antibody titers, overall effectiveness, and adverse events. While the authors are focusing on vaccination primarily in China, reports from other countries with similar products should be included in the introduction and discussion, especially as it relates to the likelihood of adverse events.
Author response
Thank you. We have cited more related references from other countries in the introduction and discussion according to your suggestions.
Reviewer 3 Report
The paper is interesting and well written. However, I suggest to improve it through a few minor revisions. First: it is useful to discuss the efficacy and safety of the vaccines in autoimmune patients. Second: a briefly discussion upon the probably seasonal vaccination against Sars-Cov2 as other vaccines always in autoimmune patients (e.g., flu, pneumococcal).
Author Response
Author response
Thank you. We added them in the discussion section. We also edited English language in the revision.